# ACTION SEMANTICS NETWORK: CONSIDERING THE EFFECTS OF ACTIONS IN MULTIAGENT SYSTEMS

**Weixun Wang**[1,*] **Tianpei Yang**[1,*] **Yong Liu**[2], **Jianye Hao**[1,3,4,†] **Xiaotian Hao**[1], **Yujing Hu**[5],
**Yingfeng Chen**[5], **Changjie Fan**[5], **Yang Gao**[2]
{wxwang, tpyang}@tju.edu.cn, lucasliunju@gmail.com, {jianye.hao, xiaotianhao}@tju.edu.cn,
{huyujing, chenyingfeng1, fanchangjie}@corp.netease.com, gaoy@nju.edu.cn
[1]College of Intelligence and Computing, Tianjin University
[2]Nanjing University
[3]Tianjin Key Lab of Machine Learning, Tianjin University
[4]Noah's Ark Lab, Huawei
[5]NetEase Fuxi AI Lab

## ABSTRACT

In multiagent systems (MASs), each agent makes individual decisions but all of them contribute globally to the system evolution. Learning in MASs is difficult since each agent's selection of actions must take place in the presence of other co-learning agents. Moreover, the environmental stochasticity and uncertainties increase exponentially with the increase in the number of agents. Previous works borrow various multiagent coordination mechanisms into deep learning architecture to facilitate multiagent coordination. However, none of them explicitly consider action semantics between agents that different actions have different influence on other agents. In this paper, we propose a novel network architecture, named Action Semantics Network (ASN), that explicitly represents such action semantics between agents. ASN characterizes different actions' influence on other agents using neural networks based on the action semantics between them. ASN can be easily combined with existing deep reinforcement learning (DRL) algorithms to boost their performance. Experimental results on StarCraft II micromanagement and Neural MMO show ASN significantly improves the performance of state-of-the-art DRL approaches compared with several network architectures.

## 1 INTRODUCTION

Deep reinforcement learning (DRL) (Sutton & Barto, 2018) has achieved a lot of success at finding optimal policies to address single-agent complex tasks (Mnih et al., 2015; Lillicrap et al., 2016; Silver et al., 2017). However, there also exist a lot of challenges in multiagent systems (MASs) since agents' behaviors are influenced by each other and the environment exhibits more stochasticity and uncertainties (Claus & Boutilier, 1998; Hu & Wellman, 1998; Bu et al., 2008; Hauwere et al., 2016).

Recently, a number of deep multiagent reinforcement learning (MARL) approaches have been proposed to address complex multiagent problems, e.g., coordination of robot swarm systems (Sosic et al., 2017) and autonomous cars (Oh et al., 2015). One major class of works incorporates various multiagent coordination mechanisms into deep multiagent learning architecture (Lowe et al., 2017; Foerster et al., 2018; Yang et al., 2018; Palmer et al., 2018). Lowe et al. (2017) proposed a centralized actor-critic architecture to address the partial observability in MASs. They also incorporate the idea of joint action learner (JAL) (Littman, 1994) to facilitate multiagent coordination. Later, Foerster et al. (2018) proposed Counterfactual Multi-Agent Policy Gradients (COMA) motivated from the difference reward mechanism (Wolpert & Tumer, 2001) to address the challenges of multiagent credit assignment. Recently, Yang et al. (2018) proposed applying mean-field theory (Stanley, 1971) to solve large-scale multiagent learning problems. More recently, Palmer et al. (2018) extended the idea of leniency (Potter & Jong, 1994; Panait et al., 2008) to deep MARL and proposed the retroactive temperature decay schedule to address stochastic rewards problems. However, all these

---

[*]Equal contribution, † corresponding author

works ignore the natural property of the action influence between agents, which we aim to exploit to facilitate multiagent coordination.

Another class of works focus on specific network structure design to address multiagent learning problems (Sunehag et al., 2018; Rashid et al., 2018; Sukhbaatar et al., 2016; Singh et al., 2019). Sunehag et al. (2018) designed a value-decomposition network (VDN) to learn an optimal linear value decomposition from the team reward signal based on the assumption that the joint action-value function for the system can be additively decomposed into value functions across agents. Later, Rashid et al. (2018) relaxed the linear assumption in VDN by assuming that the Q-values of individual agents and the global one are also monotonic, and proposed QMIX employing a network that estimates joint action-values as a complex non-linear combination of per-agent values. Recently, Zambaldi et al. (2019) proposed the relational deep RL to learn environmental entities relations. However, they considered the entity relations on the pixel-level of raw visual data, which ignores the natural property of the influence of actions between agents. Tacchetti et al. (2019) proposed a novel network architecture called Relational Forward Model (RFM) for predictive modeling in multiagent learning. RFM takes a semantic description of the state of an environment as input, and outputs either an action prediction for each agent or a prediction of the cumulative reward of an episode. However, RFM does not consider from the perspective of the influence of each action on other agents. OpenAI designed network structures to address multiagent learning problems in a famous Multiplayer Online Battle Arena (MOBA), Dota2. They used a scaled-up version of PPO (Schulman et al. (2017)), adopted the attention mechanism to compute the weight of choosing the target unit, with some of information selected from all information as input. However, this selection is not considered from the influence of each action on other agents. There are also a number of works designing network structures for multiagent communication (Sukhbaatar et al., 2016; Singh et al., 2019).

However, none of the above works explicitly leverage the fact that an agent's different actions may have different impacts on other agents, which is a natural property in MASs and should be considered in the decision-making process. In multiagent settings, each agent's action set can be naturally divided into two types: one type containing actions that affect environmental information or its private properties and the other type containing actions that directly influence other agents (i.e., their private properties). Intuitively, the estimation of performing actions with different types should be evaluated separately by explicitly considering different information. We refer to the property that different actions may have different impacts on other agents as action semantics. We can leverage the action semantics information to improve an agent's policy/Q network design toward more efficient multiagent learning.

To this end, we propose a novel network architecture, named Action Semantics Network (ASN) to characterize such action semantics for more efficient multiagent coordination. The main contributions of this paper can be summarized as follows: 1) to the best of our knowledge, we are the first to explicitly consider action semantics and design a novel network to extract it to facilitate learning in MASs; 2) ASN can be easily combined with existing DRL algorithms to boost its learning performance; 3) experimental results[*] on StarCraft II micromanagement (Samvelyan et al., 2019) and Neural MMO (Suarez et al., 2019) show our ASN leads to better performance compared with state-of-the-art approaches in terms of both convergence speed and final performance.

## 2 BACKGROUND

Stochastic games (SGs) (Littman, 1994) are a natural multiagent extension of Markov decision processes (MDPs), which models the dynamic interactions among multiple agents. Considering the fact that agents may not have access to the complete environmental information, we follow previous work's settings and model the multiagent learning problems as partially observable stochastic games (POSGs) (Hansen et al., 2004).

A *partially observable stochastic game* (POSG) is defined as a tuple $\langle \mathcal{N}, \mathcal{S}, \mathcal{A}^1, \cdots, \mathcal{A}^n, \mathcal{T}, \mathcal{R}^1, \cdots, \mathcal{R}^n, \mathcal{O}^1, \cdots, \mathcal{O}^n \rangle$, where $\mathcal{N}$ is the set of agents; $\mathcal{S}$ is the set of states; $\mathcal{A}^i$ is the set of actions available to agent $i$ (the joint action space $\mathcal{A} = \mathcal{A}^1 \times \mathcal{A}^2 \times \cdots \times \mathcal{A}^n$); $\mathcal{T}$ is the transition function

---

[*]More details can be found at `https://sites.google.com/view/iclrasn`, the source code is put on `https://github.com/MAS-anony/ASN`

that defines transition probabilities between global states: $\mathcal{S} \times \mathcal{A} \times \mathcal{S} \rightarrow [0,1]$; $\mathcal{R}^i$ is the reward function for agent $i$: $\mathcal{S} \times \mathcal{A} \rightarrow \mathbb{R}$ and $\mathcal{O}^i$ is the set of observations for agent $i$.

Note that a state $s \in \mathcal{S}$ describes the environmental information and the possible configurations of all agents, while each agent $i$ draws a private observation $o^i$ correlated with the state: $\mathcal{S} \mapsto \mathcal{O}^i$, e.g., an agent's observation includes the agent's private information and the relative distance between itself and other agents. Formally, an observation of agent $i$ at step $t$ can be constructed as follows: $o_t^i = \{o_t^{i,env}, m_t^i, o_t^{i,1}, \cdots, o_t^{i,i-1}, o_t^{i,i+1}, \cdots, o_t^{i,n}\}$, where $o_t^{i,env}$ is the observed environmental information, $m_t^i$ is the private property of agent $i$ (e.g., in robotics, $m_t^i$ includes agent $i$'s location, the battery power and the healthy status of each component) and the rest are the observations of agent $i$ on other agents (e.g., in robotics, $o_t^{i,i-1}$ includes the relative location, the exterior of agent $i-1$ that agent $i$ observes). An policy $\pi_i: \mathcal{O}^i \times \mathcal{A}^i \rightarrow [0;1]$ specifies the probability distribution over the action space of agent $i$. The goal of agent $i$ is to learn a policy $\pi_i$ that maximizes the expected return with a discount factor $\gamma$: $J = \mathbb{E}_{\pi_i} \left[ \sum_{t=0}^{\infty} \gamma^t r_t^i \right]$.

# 3 THE ACTION SEMANTICS NETWORK ARCHITECTURE

## 3.1 MOTIVATION

In MASs, multiple agents interact with the environment simultaneously which increases the environmental stochasticity and uncertainties, making it difficult to learn a consistent globally optimal policy for each agent. A number of Deep Multiagent Reinforcement Learning (MARL) approaches have been proposed to address such complex problems in MASs by either incorporating various multiagent coordination mechanisms into deep multiagent learning architecture (Foerster et al., 2018; Yang et al., 2018; Palmer et al., 2018) or designing specialized network structures to facilitate multiagent learning (Sunehag et al., 2018; Rashid et al., 2018; Sukhbaatar et al., 2016). However, none of them explicitly consider extracting action semantics, which we believe is a critical factor that we can leverage to facilitate coordination in multiagent settings. Specifically, each agent's action set can be naturally classified into two types: one type containing actions that directly affect environmental information or its private properties and the other type of actions directly influence other agents. Therefore, if an agent's action directly influences one of the other agents, the value of performing this action should be explicitly dependent more on the agent's observation for the environment and the information of the agent to be influenced by this action, while any additional information (e.g., part of the agent's observation for other agents) is irrelevant and may add noise. We refer to the property that different actions may have different impacts on other agents as action semantics.

However, previous works usually use all available information for estimating the value of all actions, which can be quite inefficient. To this end, we propose a new network architecture called Action Semantics Network (ASN) that explicitly considers action semantics between agents to improve the estimation accuracy over different actions. Instead of inputting an agent's total observation into one network, ASN consists of several sub-modules that take different parts of the agent's observation as input according to the semantics of actions. In this way, ASN can effectively avoid the negative influence of the irrelevant information, and thus provide a more accurate estimation of performing each action. Besides, ASN is general and can be incorporated into existing deep MARL frameworks to improve the performance of existing DRL algorithms. In the next section, we will describe the ASN structure in detail.

## 3.2 ASN

Considering the semantic difference of different actions, we classify an agent's action set $\mathcal{A}^i$ of agent $i$ into two subsets: $\mathcal{A}_{in}^i$ and $\mathcal{A}_{out}^i$. $\mathcal{A}_{in}^i$ contains actions that affect the environmental information or its private properties and do not influence other agents directly, e.g., moving to different destinations would only affect its own location information. $\mathcal{A}_{out}^i$ corresponds to those actions that directly influence some of other agents, e.g., attack agent $j$ in competitive settings or communicate with agent $j$ in cooperative settings.

Following the above classification, the proposed network architecture, ASN, explicitly considers the different influence of an agent's actions on other agents by dividing the network into different sub-modules, each of which takes different parts of the agent's observation as input according to

the semantics of actions (shown in Figure 1). Considering an agent $i$ and $n-1$ agents in its neighborhood, ASN decouples agent $i$'s network into $n$ sub-modules as follows. The first one shown in Figure 1 (left side $O2A^i$) contains a network $O2E^i$ which is used to generate the observation embedding $e^i$ given the full observation $o^i_t$ of agent $i$ as input, and a network $E2A^i$ (embedding to action) which generates the values of all action in $\mathcal{A}^i_{in}$ as output. The rest of $n-1$ sub-modules ($O2A^{i,j}, j \in \mathcal{N}, j \neq i$) are used to estimate the values of those actions in $\mathcal{A}^i_{out}$ related with each influenced agent, composed of $n-1$ networks ($O2E^{i,j}, j \in \mathcal{N}, j \neq i$) which are responsible for determining the observation embeddings related with each influenced agent, denoted as $e^{i,j}$. Each of $n-1$ sub-modules $O2A^{i,j}$ only takes a part of agent $i$'s observation related with one neighbor agent $j$, $o^{i,j}_t$ as input.

For value-based RL methods, at each step $t$, the evaluation of executing each action $a^i_t \in \mathcal{A}^i_{in}$ is $Q(o^i_t, a^i_t) = fa(e^i_t, a^i_t)$, where $fa(e^i_t, a^i_t)$ is one of the outputs of the $E2A^i$ network corresponding to $a^i_t$. To evaluate the performance of executing an action $a^{i,j}_t \in \mathcal{A}^i_{out}$ on another agent $j$, ASN combines these two embeddings $e^i_t$ and $e^{i,j}_t$ using a pairwise interaction function $\mathcal{M}$ (e.g., inner product):

$$Q(o^i_t, a^{i,j}_t) = \mathcal{M}(e^i_t, e^{i,j}_t) \quad (1)$$

then agent $i$ selects the action $a^i_t = \arg\max_{a^i_t \in A^i}\{Q(o^i_t, a^i_t)\}$ with certain exploration $\epsilon$.

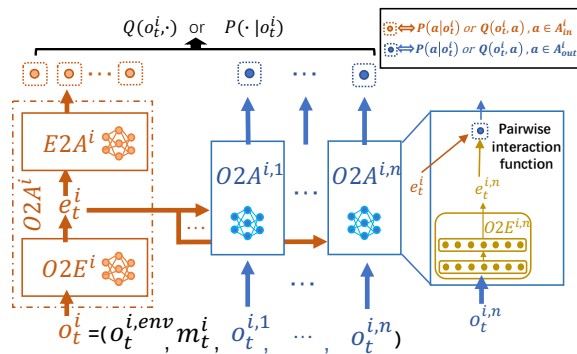

Figure 1: ASN of agent $i$ contains $n$ sub-modules: $O2A^i, O2A^{i,1}, \cdots, O2A^{i,i-1}, O2A^{i,i+1}, \cdots, O2A^{i,n}$, each of which takes different parts of the agent's observation as input.

Similarly, if the policy is directly optimized through policy-based RL methods, the probability of choosing each action is proportional to the output of each sub-module: $\pi(a^i_t|o^i_t) \propto \exp(fa(e^i_t, a^i_t)), \pi(a^{i,j}_t|o^i_t) \propto \exp(\mathcal{M}(e^i_t, e^{i,j}_t))$. Then agent $i$ selects an action following $\pi^i$:

$$\pi(a^i_t|o^i_t) = \frac{\exp(fa(e^i_t, a^i_t))}{Z^{\pi_i}(o^i_t)} , \pi(a^{i,j}_t|o^i_t) = \frac{\exp(\mathcal{M}(e^i_t, e^{i,j}_t))}{Z^{\pi_i}(o^i_t)} \quad (2)$$

where $Z^{\pi_i}(o^i_t)$ is the partition function that normalizes the distribution. Note that we only consider the case that an action $a^{i,j}$ directly influences one particular agent $j$. In general, there may exist multiple actions directly influencing one particular agent and how to extend our ASN will be introduced in Section 3.3(Multi-action ASN).

### 3.3 ASN-MARL

Next, we describe how ASN can be incorporated into existing deep MARL, which can be classified into two paradigms: Independent Learner (IL) (Mnih et al., 2015; Schulman et al., 2017) and Joint Action Learner (JAL) (Lowe et al., 2017; Rashid et al., 2018; Foerster et al., 2018). IL applies a single-agent learning algorithm to a multiagent domain to treat other agents as part of the environment. In contrast, JALs observe the actions of other agents, and optimize the policy for each joint action. Following the above two paradigms, we propose two classes of ASN-based MARL: ASN-IL and ASN-JAL. For ASN-IL, we focus on the case of combing ASN with PPO (Schulman et al., 2017), a popular single-agent policy-based RL. The way ASN combines with other single-agent RL is similar. In contrast, ASN-JAL describes existing deep MARL approaches combined with ASN, e.g., QMIX (Rashid et al., 2018) and VDN (Sunehag et al., 2018).

**ASN-PPO** For each agent $i$ equipped with a policy network parameterized by $\theta^i$, ASN-PPO replaces the vanilla policy network architecture with ASN and optimizes the policy following PPO.

Generally, IL ignores the existence of other agents, thus, for each agent $i$, the expected return $J(\theta^i)$ is optimized using the policy gradient theorem: $\nabla_{\theta^i} J(\theta^i) = \mathbb{E}_t \left[ \nabla_{\theta^i} \log \pi_{\theta^i}(a^i_t|o^i_t) A_t(o^i_t, a^i_t) \right]$,

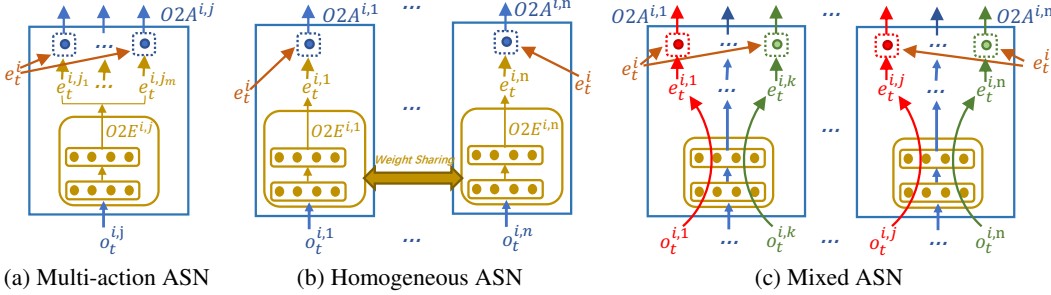

(a) Multi-action ASN      (b) Homogeneous ASN      (c) Mixed ASN

Figure 3: Different variants of ASN. Here we only present the right part of ASN (excluding the left part $O2A^i$ of ASN) as different variants.

where $A_t$ is the advantage function at timestep $t$. PPO uses constraints and advantage estimation to reformulate the optimization problem as:

$$\max_{\theta^i} \mathbb{E}_t \left[ r_t(\theta^i) A_t(o_t^i, a_t^i) \right] \tag{3}$$

where $r_t(\theta^i)$ is the probability ratio $\frac{\pi_{\theta^i}(a_t^i|o_t^i)}{\pi_{\theta_{old}^i}(a_t^i|o_t^i)}$, $\theta_{old}^i$ is the policy parameters before the update. Then in ASN-PPO, $r_t(\theta^i)$ can be rewritten as follows by substituting Equation 2:

$$r_t(\theta^i) = \begin{cases} \frac{\exp(fa(\boldsymbol{e}_t^i, a_t^i; \theta^i))}{\exp(fa(\boldsymbol{e}_t^i, a_t^i; \theta_{old}^i))} \frac{Z^{\pi^i}(o_t^i; \theta_{old}^i)}{Z^{\pi^i}(o_t^i; \theta^i)} & \text{if } a_t^i \in \mathcal{A}_{in}^i \\ \frac{\exp(\mathcal{M}(\boldsymbol{e}_t^i, \boldsymbol{e}_t^{i,j}; \theta^i))}{\exp(\mathcal{M}(\boldsymbol{e}_t^i, \boldsymbol{e}_t^{i,j}; \theta_{old}^i))} \frac{Z^{\pi^i}(o_t^i; \theta_{old}^i)}{Z^{\pi^i}(o_t^i; \theta^i)} & \text{if } a_t^i \in \mathcal{A}_{out}^i \end{cases} \tag{4}$$

Lastly, ASN-PPO maximizes the objective (Equation 3) following PPO during each iteration.

**ASN-QMIX** The way ASN combines with deep MARL algorithms is similar and we use QMIX (Rashid et al., 2018) as an example to present. Figure 2 illustrates the ASN-QMIX network structure, where for each agent $i$, ASN-QMIX replaces the vanilla Q-network architecture with ASN. At each step $t$, the individual Q-function $Q(o_t^i, a_t^i)$ is first calculated following Section 3.2 and then input into the mixing network. The mixing network mixes the output of all agents' networks monotonically and produces the joint action-value function $Q_{tot}(s_t, a_t)$. The weights of

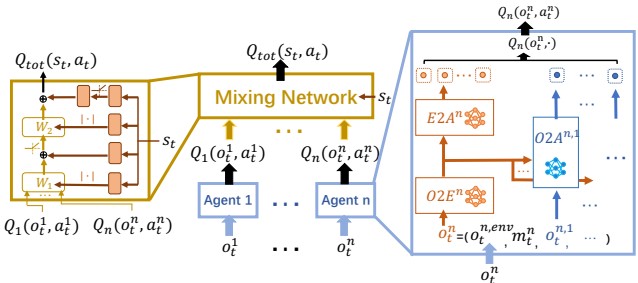

Figure 2: QMIX-ASN contains one mixing network and $n$ networks of all agents, and each agent network follows the ASN architecture.

the mixing network are restricted to be non-negative and produced by separate hypernetworks, each of which takes state $s_t$ as input and generates the weights of one layer of the mixing network. Finally, ASN-QMIX is trained to minimize the loss: $L(\theta) = \sum_{b=1}^{B} \left[ (y_b^{tot} - Q_{tot}(s, \boldsymbol{a}; \theta))^2 \right]$, where $B$ is the batch size of transitions, $y_t^{tot} = r_t + \gamma \max_{\boldsymbol{a}'} Q_{tot}(s', \boldsymbol{a}'; \theta^-)$, and $\theta^-$ are the parameters of the target network as in DQN (Mnih et al., 2015).

**Multi-action ASN** The general case in MASs is that an agent may have multiple actions which can directly influence another agent, e.g., a router can send packages with different size to one of its neighbors, a soldier can select different weapons to attack enemies and cause different damages. To address this, we extend the basic ASN to a generalized version, named Multi-action ASN (shown in Figure 3(a)), that takes $o^{i,j}$ as input, and produces a number of embeddings $\boldsymbol{e}^{i,j_1}, \cdots, \boldsymbol{e}^{i,j_m}$, where $m$ is the number of actions that directly influences agent $j$. After that, multi-action ASN calculates the estimation of performing each action, which uses a pairwise interaction function $\mathcal{M}$ to combine the two embeddings $\boldsymbol{e}^{i,j_{k,k \in [1,m]}}$ and $\boldsymbol{e}^i$ following Equation (1).

**Parameter-sharing between sub-modules**     Parameter-sharing (PS) mechanism is widely used in MARL. If agents are homogeneous, their policy networks can be trained more efficiently using PS which greatly reduces the training complexity (Gupta et al., 2017). Recent work (Rashid et al., 2018) also incorporates PS on heterogeneous agents by adding extra information to identify agent type. Following previous work, here we incorporate PS to enable parameter-sharing between different sub-modules of ASN. The basic ASN (Figure 1) for agent $i$ contains a number of sub-modules $O2A^{i,j}$, each of which takes $o^{i,j}$ as input. In this way, if an action $a_t^{i,j} \in \mathcal{A}_{out}^i$ has a direct impact on any of another agent $j$, the number of sub-modules is equal to the number of other agents. The training of basic ASN is inefficient since the number of sub-modules is increasing with the increase in the number of agents. If the other agents that agent $i$ can directly influence are homogeneous, the sub-module parameters can be shared across those agents. Thus, in a homogeneous MAS, all influencing agents can share one sub-module (shown in Figure 3 (b)); in a MAS that contains several types of agents, each type of agents can share one sub-module (Mixed ASN in Figure 3 (c)). Note that the basic ASN can be seen as the simplest case that designs a sub-module for each influencing agent without PS.

## 4 SIMULATIONS

We evaluate the performance of ASN compared with different network structures including the vanilla network (i.e., aggregate all information and input into one single network), the dueling network (Wang et al., 2016), the attention network that expects to learn which information should be focused on more automatically (i.e., adds an additional hidden layer to compute the weights of the input and then generate an element-wise product to input into the next layer) and entity-attention network (i.e., instead of computing attention weight for each dimension of the input, the weight is computed for each entity/agent) under various DRL approaches. Other network architectures as we mentioned before are not comparable here since they are orthogonal to our ASN. Our test domains include StarCraft II micromanagement (Samvelyan et al., 2019) and Massively Multiplayer Online Role-Playing Games (Neural MMO) (Suarez et al., 2019). The details of neural network structures and parameter settings are in the appendix.

### 4.1 STARCRAFT II

StarCraft II is a real-time strategic game with one or more humans competing against each other or a built-in game AI. Here we focus on a decentralized multiagent control that each of the learning agents controls an individual army entity. At each step, each agent observes the local game state which consists of the following information for all units in its field of view: relative distance between other units, the position and unit type (detailed in the appendix) and selects one of the following actions: move north, south, east or west, attack one of its enemies, stop and the null action. Agents belonging to the same side receive the same joint reward at each time step that equals to the total damage on the enemy units. Agents also receive a joint reward of 10 points after killing each enemy, and 200 points after killing all enemies. The game ends when all agents on one side die or the time exceeds a fixed period. Note that previous works (Foerster et al., 2018; Rashid et al., 2018; Samvelyan et al., 2019) reduce the learning complexity by manually adding a rule that forbids each agent to select an invalid action, e.g., attack an opponent that beyond the attack range and move beyond the grid border. We relax this setting since it requires prior knowledge, which is hard to obtain in the real world. We are interested in evaluating whether these rules can be learned automatically through end-to-end training as well. Thus, the following results are based on the setting that each agent can select an action that causes an invalid effect, and in result, the agent will standstill at the current time step. We also evaluate ASN following previous settings (adding the manual rule in StarCraft II that forbidding the invalid actions) and ASN still achieves better performance which can be found in the appendix.

In StarCraft II 8m map (8 Marines vs 8 Marines), each agent is homogeneous to each other, so we adopt **homogeneous** ASN to evaluate whether it can efficiently characterize action semantics between two agents. Figure 4(a), (b) and (c) show the performance of ASN on an 8m map compared with vanilla, dueling, attention and entity-attention networks under different DRL algorithms (IQL, QMIX, VDN). We can see that ASN performs best among all of the network structures in terms of both convergence rate and average win rates. By taking different observation information as the

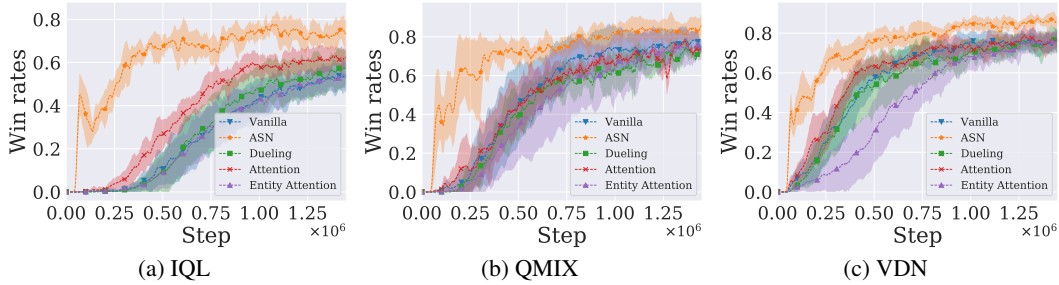

Figure 4: Win rates of various methods on the StarCraft II 8m map.

input of different sub-modules, ASN enables an agent to learn the right timing to attack different opponents to maximize its total damage on opponents. In contrast, existing network architectures simply input all information into one network, thus an agent cannot distinguish the difference of effects that different actions may have on the opponents and may choose the suboptimal opponent to attack, thus resulting in lower performance than ASN. Attention network performs better than vanilla and dueling when combined with IQL, while both of them show very similar performance with the vanilla network when combined with QMIX and VDN. However, entity-attention performs worst since it is hard to figure out the useful information for each entity when input all information into one network initially. Since the performance difference of other network architecture is marginal, we only present results of ASN-QMIX compared with the vanilla network under QMIX (denoted as vanilla-QMIX) in the following sections.

Next, we consider a more complex scenario: StarCraft II 2S3Z (2 Stalkers and 3 Zealots vs 2 Stalkers and 3 Zealots) which contains two **heterogeneous** groups, each agent inside one group are homogeneous and can evaluate the performance of **Mixed ASN** compared with vanilla-QMIX. From Figure 5(a) we can observe that Mixed ASN-QMIX perform better than vanilla-QMIX. The reason is that ASN efficiently identifies action semantics between each type of two agents, thus it selects more proper attack options each time and achieves better performance last vanilla-QMIX.

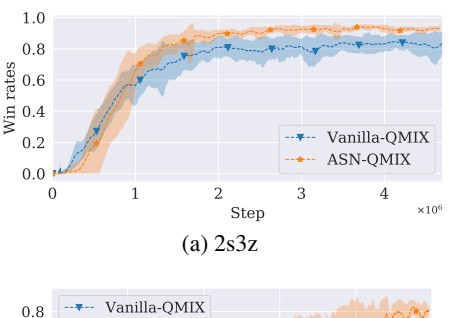

(a) 2s3z

**Is ASN still effective on large-scale scenarios?** We further test on a **large-scale** agent space on a 15m map. Figure 5 (b) depicts the dynamics of the average win rates of ASN-QMIX and vanilla-QMIX. We can see that ASN-QMIX quickly learns the average win rates of approximately 80 %, while vanilla-QMIX fails, with the average win rates of approximately only 20 %. From Figure 4 (b) and 5 (b) we can find that with the increase of the agent number, the margin becomes larger between two methods. Intuitively, ASN enables an agent to explicitly consider more numbers of other agents' information with a larger agent size. However, for the vanilla network, it is more difficult to identify the action influence on other agents from a larger amount of mixed information, which results in lower average win rates than ASN. An interesting observation for vanilla-QMIX is that they will learn to run away to avoid all being killed, and testing videos can be found in our anonymous website*.

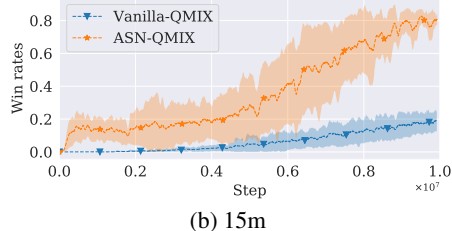

(b) 15m

Figure 5: Win rates of ASN-QMIX and vanilla-QMIX on different SC II maps.

Table 1: PCT of choosing a valid action for ASN-QMIX and vanilla-QMIX.

|  | ASN | Vanilla |
|---|---|---|
| PCT | $71.9 \pm 0.15\%$ | $44.3 \pm 0.11\%$ |

**Can ASN recognize the influence of different actions?** Table 1 presents the average percentages of choosing a valid action for ASN-QMIX and vanilla-QMIX on a 15m map. Note that we

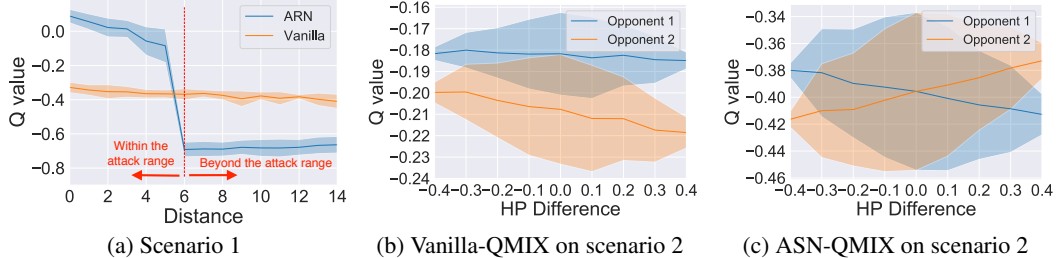

Figure 6: The attack action's Q-values of ASN and vanilla under different circumstances.

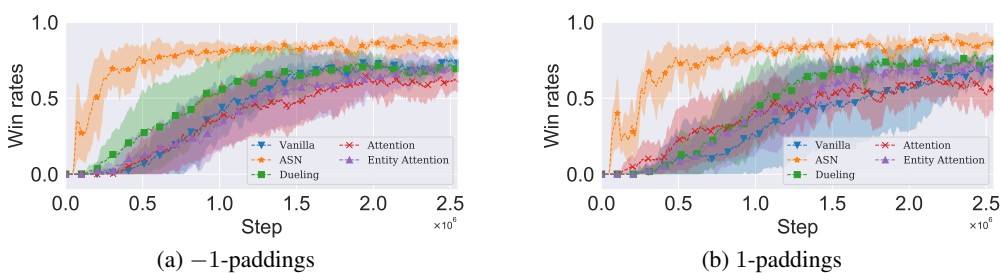

Figure 7: Win rates on SC II 8m map when replacing 0-paddings with −1-paddings and 1-paddings.

remove the manually added rule (which prevents selecting any invalid action), and agents would probably select the invalid action and standstill, which increases the learning difficulties. We can see that ASN-QMIX achieves an average percentage of approximately 71.9% for choosing a valid action. However, vanilla-QMIX only achieves an average percentage of approximately 44.3%. This phenomenon confirms that ASN effectively exploits action semantics between agents and enables agents to learn which action can be chosen at each time step, facilitating more robust learning, even in large-scale MASs.

**Can ASN effectively improve the estimation accuracy of actions?**     We investigate whether ASN can efficiently characterize the action semantics and facilitate multiagent coordination. To make the analysis more clear, we test the model learned on a 15m map on two illustrating scenarios: 1) the one-on-one combat scenario that the distance between two agents is dynamically changing; 2) the one Marine vs two Marines scenario that the HPs (Hit Points) of two opponents are dynamically different. Figure 6(a) shows the dynamics of the attack action's Q-value with the distance change of the ASN agent and its opponent. We can observe that the Q-value of the action that the ASN agent attacking its opponent decreases as the distance of the agent and its opponent increases, and stabilizes when the distance exceeds the attack range. However, the vanilla agent keeps the Q-value of the attack action nearly unchanged. This indicates that ASN can automatically learn the information of when an action is valid and behave appropriately, while the vanilla agent has to rely on manually added rules to avoid choosing invalid actions. Figure 6 (b) and (c) shows the dynamics of the attack action's Q-value of ASN agent and vanilla agent with the HPs difference of two opponents changing (i.e., the HP difference equals to the HP of opponent 1 minus the HP of opponent 2). We can see that the ASN agent holds a higher Q-value of attacking opponent 1 when opponent 1's HP is lower than opponent 2 and vice versa. The symmetric curve of ASN is due to the fact that the state description of two opponents is very similar in this scenario. However, the vanilla agent always keeps a higher attack action's Q-value on Opponent 1 than on Opponent 2, which means it always selects to attack Opponent 1. These results indicate that ASN can effectively exploit the action semantics between agents and improves the estimation accuracy on different actions, thus facilitates robust learning among agents.

**Does ASN exploits the 0-padding information?**     When one of the army units dies or some units are beyond the range of vision, one common practice is to use 0-paddings as the input for the observation of the died army unit. In this section, we provide an ablation study on whether ASN design

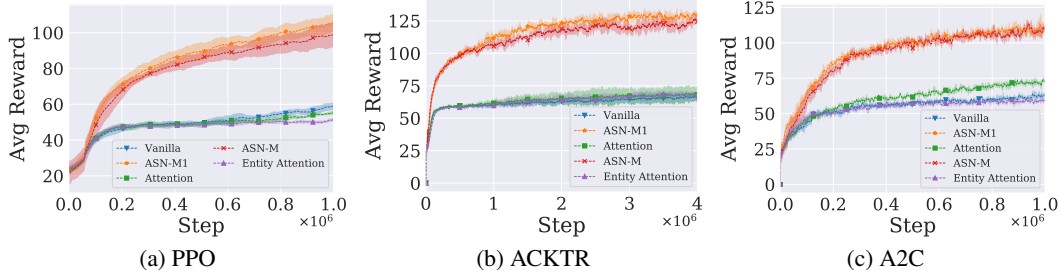

(a) PPO                    (b) ACKTR                    (c) A2C

Figure 9: Average rewards of various methods on Neural MMO.

exploits the 0-padding information. Figure 7 shows the win rates of various network architectures combined with QMIX when using 1-paddings and −1-paddings as the input for the observation of the died army unit. We can see that ASN still performs best among all network architectures in terms of both convergence speed and final win rates. This indicates that ASN effectively extracts the action semantics between agents, instead of benefiting from the particular settings of 0-paddings.

## 4.2 NEURAL MMO

The Neural MMO (Suarez et al., 2019) is a massively multiagent environment that defines combat systems for a large number of agents. Figure 8 illustrates a simple Neural MMO scene with two groups of agents on a $10 \times 10$ tile. Each group contains 3 agents, each of which starts at any of the tiles, with HP = 100. At each step, each agent loses one unit of HP, observes local game state (detailed in the appendix) and decides on an action, i.e., moves one tile (up, right, left, down and stop) or makes

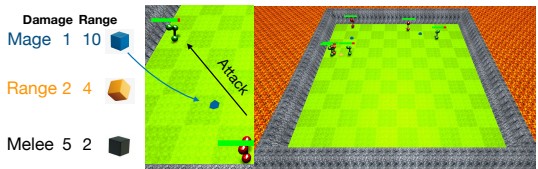

Figure 8: An illustration of Neural MMO that contains two armies (red and green).

an attack using any of three attack options (shown in the left part in Figure 8: "Melee" with the attack distance is 2, the amount of damage is 5; "Range" with the attack distance is 4, the amount of damage is 2; "Mage" with the attack distance is 10, the amount of damage is 1). Each action that causes an invalid effect (e.g., attack an opponent that beyond the attack range and move beyond the grid border) would make the agent standstill. Each agent gets a penalty of −0.1 if the attack fails. The game ends when all agents in one group die, and agents belonging to the same group receive a joint reward, which is the difference of the total HPs between itself and its opposite side.

In Neural MMO, an agent can attack one of its opponent using one of three different attack options, which can be used to evaluate whether **multi-action** ASN can efficiently identify the multiple action semantics between agents. Here we adopt two kinds of multi-action ASN: ASN-M1 that shares parameters of the first neural network layer across three attack actions on one enemy (as shown in Figure 3(a)); and ASN-M that does not share. Figure 9(a), (b) and (c) present the performance of multi-action AS-N on Neural MMO compared with vanilla, attention and entity-attention networks under different IL methods (PPO, ACKTR (Wu et al., 2017) and A2C (Mnih et al., 2016)). We can observe that ASN performs best under all three IL approaches in terms of average rewards. This is because ASN can learn to choose appropriate actions against other agents at different time steps to maximize the damage on others. However, the vanilla network

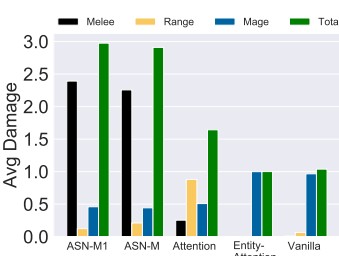

Figure 10: The average damage of choosing each attack when distance $d_{ij} \leq 2$ under A2C.

just mixes all information together which makes it difficult to identify and take advantage of the action semantics between agents, thus it achieves lower performance than ASN. Since the information is mixed initially, although the attention and entity-attention networks try to learn which information

should be focused on more, it is hard to distinguish which part of the information is more useful, thus achieving lower performance than ASN.

**Can ASN recognize the best actions from multiple ones?** We further investigate whether ASN can efficiently exploit different action semantics between agents and enable an agent to identify the best attack option (i.e., an attack that causes the most damage) with the distance between the agent and its opponent changing. Figure 10 shows the average attack damage of each attack option in Neural MMO when the distance between agent $i$ and its opponent $j$ is less than or equal to 2 ($d_{ij} \leq$ 2). The best attack option is "Melee" within this distance range since it causes the maximum damage among three attacks. We can see that both ASN-M1 agent and ASN-M cause higher total damage than other methods, and ASN-M1 agent causes the highest total damage on average. However, the attention network only causes average total damage of approximately 1.5, the entity-attention and vanilla network only cause average total damage of approximately 1.0 due to the lower probability of selecting the best attack action "Melee". This is because two kinds of ASN have a larger probability to select the best attach option "Melee" than other two networks, thus causing larger total damage. Similar results on other distance ranges ($d_{i,j} \leq 4$ , $d_{i,j} \leq 10$) can be found in the appendix that ASN always causes higher total damage than other networks.

## 5 CONCLUSION AND FUTURE WORK

We propose a new network architecture, ASN, to facilitate more efficient multiagent learning by explicitly investigating the semantics of actions between agents. To the best of our knowledge, ASN is the first to explicitly characterize the action semantics in MASs, which can be easily combined with various multiagent DRL algorithms to boost the learning performance. ASN greatly improves the performance of state-of-the-art DRL methods compared with a number of network architectures. In this paper, we only consider the direct action influence between any of two agents. As future work, it is worth investigating how to model the action semantics among more than two agents. Another interesting direction is to consider the action semantics between agents in continuous action spaces.

## 6 ACKNOWLEDGEMENTS

This work is supported by the National Natural Science Foundation of China (Grant Nos.: 61702362, U1836214, 61432008) and Science and Technology Innovation 2030 - "New Generation Artificial Intelligence" Major Project No.(2018AAA0100905).

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

## A APPENDIX

### I. StarCraft II

**State Description** In StarCraft II, we follow the settings of previous works (Rashid et al., 2018; Samvelyan et al., 2019). The local observation of each agent is drawn within their field of view, which encompasses the circular area of the map surrounding units and has a radius equal to the sight range. Each agent receives as input a vector consisting of the following features for all units in its field of view (both allied and enemy): distance, relative x, relative y, and unit type. More details can be found at `https://github.com/MAS-anony/ASN` or `https://github.com/oxwhirl/smac`.

**Network Structure**

The details of different network structures for StarCraft II are shown in Figure 11. The vanilla network (Figure 11(a)) of each agent $i$ contains two fully-connected hidden layers with 64 units and one GRU layer with 64 units, taking $o_t^i$ as input. The output layer is a fully-connected layer outputs the Q-values of each action. The attention network (Figure 11(b)) of each agent $i$ contains two isolated fully-connected layers with 64 units, taking $o_t^i$ as input and computing the standard attention value for each dimension of the input. The following hidden layer is a GRU with 64 units. The output contains the Q-values of each action. The entity-attention network (Figure 11(c)) is similar to that in Figure 11(b), except that the attention weight is calculated on each $o_t^{i,j}$. The dueling network (Figure 11(d)) is the same as vanilla except for the output layer that outputs the advantages of each action and also the state value. Our homogeneous ASN (Figure 11(e)) of each agent $i$ contains two sub-modules, one is the $O2A^i$ which contains two fully-connected layers with 32 units, taking $o_t^i$ as input, following with a GRU layer with 32 units; the other is a parameter-sharing sub-module which contains two fully-connected layers with 32 units, taking each $o_t^{i,j}$ as input, following with a GRU layer with 32 units; the output layer outputs the Q-values of each action.

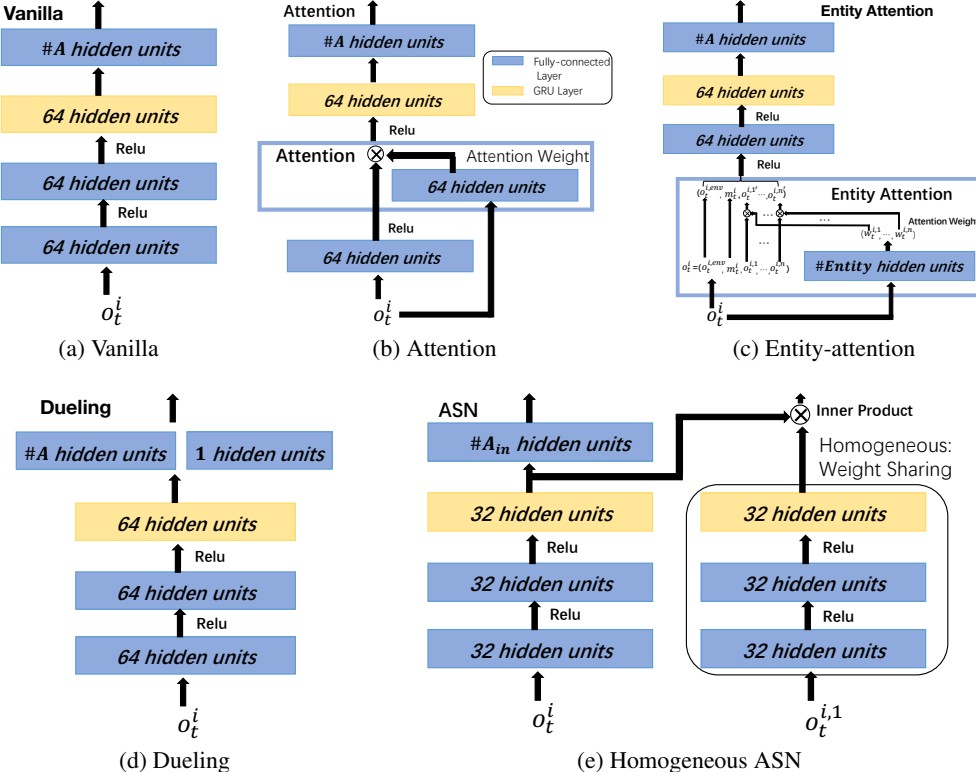

Figure 11: Various network structures on a StartCraft II 8m map.

Table 2: Hyperparameter settings for StarCraft II.

| Hyperparameter | Value |
|---|---|
| Batch-size | 32 |
| Replay memory size | 5000 |
| Discount factor($\gamma$) | 0.99 |
| Optimizer | RMSProp |
| Learning rate | $5e-4$ |
| $\alpha$ | 0.99 |
| $e$ | $1e-5$ |
| Gradient-norm-clip | 10 |
| Action-selector | $\epsilon$-greedy |
| $\epsilon$-start | 1.0 |
| $\epsilon$-finish | 0.05 |
| $\epsilon$-anneal-time | 50000 step |
| target-update-interval | 200 |

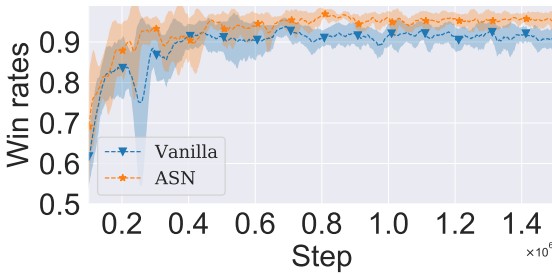

Figure 12: Win rates of ASN-QMIX and vanilla-QMIX under 5m StarCraft II map.

**Parameter Settings**

Here we provide the hyperparameters for StarCraft II [†] shown in Table 2.

**Experimental results**

The following results present the performance of ASN-QMIX and vanilla-QMIX under different StarCraft II maps with adding the manual rule (forbids the agent to choose the invalid actions).

**II. Neural MMO**

**State Description**

In a 10x10 tile (where each tile can be set as different kinds, e.g., rocks, grass), there are two teams of agents (green and red), each of which has 3 agents. At the beginning of each episode, each agent appears on any of the 10x10 tiles. The observation of an agent is in the form of a 43-dimensional vector, in which the first 8 dimensions are: time to live, HP, remaining foods (set 0), remaining water (set 0), current position (x and y), the amount of damage suffered, frozen state (1 or 0); the rest of 35 dimensions are divided equally to describe the other 5 agents' information. The first 14 dimensions describe the information of 2 teammates, following with the description of 3 opponents' information. Each observed agent's information includes the relative position(x and y), whether it is a teammate(1 or 0), HP, remaining foods, remaining water, and the frozen state.

---

[†]More details can be found at `https://github.com/MAS-anony/ASN`

Each agent chooses an action from a set of 14 discrete actions: stop, move left, right, up or down, and three different attacks against one of its opponent ("Melee" with the attack distance is 2, the amount of damage is 5; "Range" with the attack distance is 4, the amount of damage is 2; "Mage" with the attack distance is 10, the amount of damage is 1).

Each agent gets a penalty of $-0.1$ if the attack fails. They get a $-0.01$ reward for each tick and a $-10$ penalty for being killed. The game ends when a group of agents dies or the time exceeds a fixed period, and agents belonging to the same group receive the same reward, which is the difference of the total number of HPs between itself and its opposite side.

**Network Structure**

The details of vanilla, attention, entity-attention networks for Neural MMO are shown in Figure 13(a-c) which contains an actor network, and a critic network. All actors are similar to those for StarCraft II in Figure 11, except that the GRU layer is excluded and the output is the logic probability of choosing each action. All critics are the same as shown in Figure 13(a). Since in Neural MMO, each agent has multiple actions that have direct influence on each other agent, i.e., three kinds of attack options, we test two kinds of ASN variants: one (Figure 13(d)) is the Multi-action ASN we mentioned in the previous section that shares the first layer parameters among multiple actions; the other (Figure 13(e)) is the basic homogeneous ASN that does not share the first layer parameters among multiple actions.

**Parameter Settings**

Here we provide the hyperparameters for Neural MMO shown in Table 3.

**Experimental results**

The above results present the average attack damage of each attack option under the different distance ranges between the agent and its opponent.

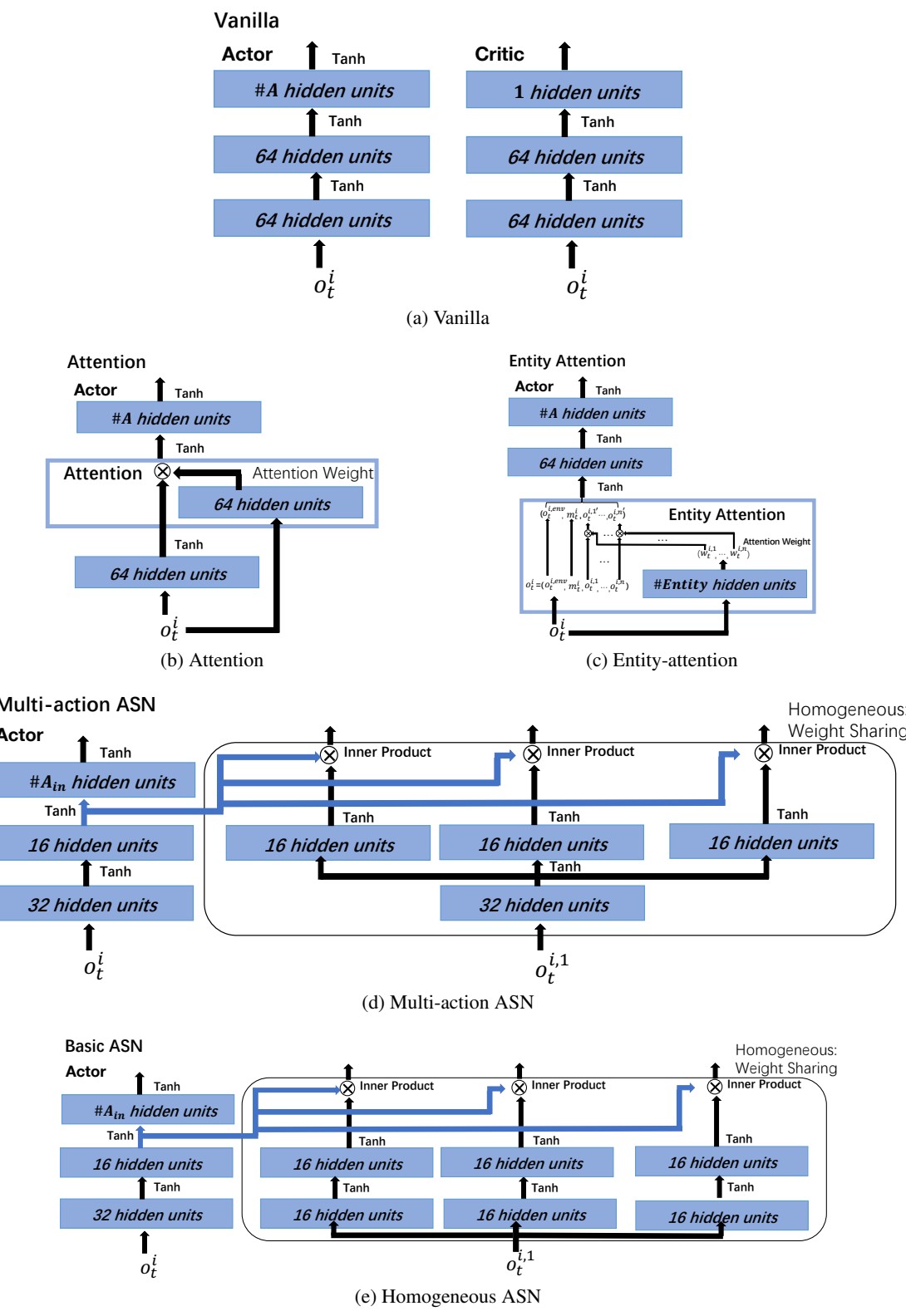

Figure 13: Various network structures on Neural MMO.

Table 3: Parameters of all algorithms.

(a) PPO

| Hyperparameter | Value |
|---|---|
| Number of processes | 1 |
| Discount factor($\gamma$) | 0.99 |
| Optimizer | Adam |
| Learning rate | $7e-4$ |
| $e$ | $1e-5$ |
| Entropy term coefficient | 1e-2 |
| Value loss coefficient | 0.5 |
| Actor loss coefficient | 1 |

(b) A2C

| Hyperparameter | Value |
|---|---|
| Number of processes | 5 |
| Discount factor($\gamma$) | 0.99 |
| Optimizer | RMSProp |
| Learning rate | $7e-4$ |
| $\alpha$ | 0.99 |
| $e$ | $1e-5$ |
| Gradient-norm-clip | 0.5 |
| Entropy term coefficient | 1e-2 |
| Value loss coefficient | 0.5 |
| Actor loss coefficient | 1 |

(c) ACKTR

| Hyperparameter | Value |
|---|---|
| Number of processes | 5 |
| Discount factor($\gamma$) | 0.99 |
| Optimizer | KFACOptimizer |
| Learning rate | 0.25 |
| Momentum | 0.9 |
| Stat decay | 0.99 |
| KL clip | 1e-3 |
| Damping | 1e-2 |
| Weight decay | 0 |
| Entropy term coefficient | 1e-2 |
| Value loss coefficient | 0.5 |
| Actor loss coefficient | 1 |

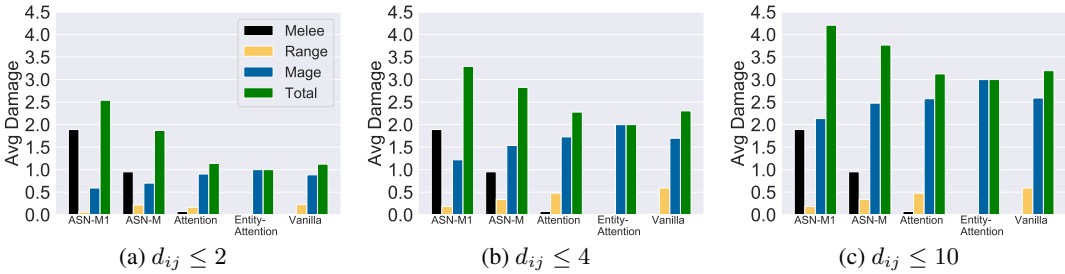

Figure 14: The average probabilities of choosing each attack under different distance $d_{ij}$ for various network architectures combined with PPO in Neural MMO.

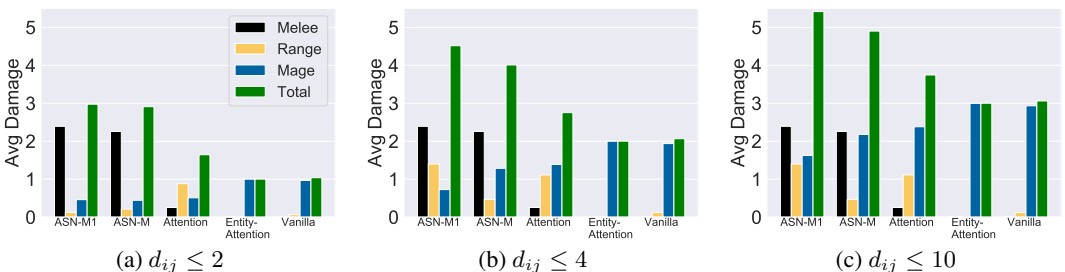

Figure 15: The average probabilities of choosing each attack under different distance $d_{ij}$ for various network architectures combined with A2C in Neural MMO.

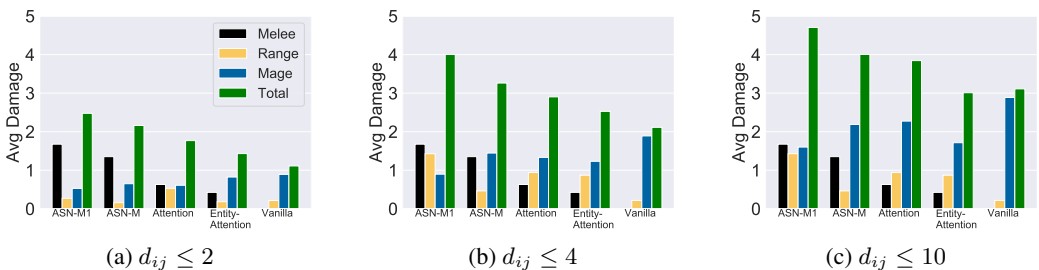

Figure 16: The average probabilities of choosing each attack under different distance $d_{ij}$ for various network architectures combined with ACKTR in Neural MMO.

