# OpenReview forum: "Action Semantics Network: Considering the Effects of Actions in Multiagent Systems"
_ICLR.cc/2020/Conference — Accept (Poster)_

### Official Review · AnonReviewer2 · 2019-10-22
**Official Blind Review #2**

**Rating:** 6

**Review:**

This paper proposes a neural network architecture that provides an agent-agent based embeddings that are used for actions that directly affect specific agent. Proposed architectural choice exploits (implicitly assumed) independence of some actions wrt. observations of agents that it is not directly affecting. Authors perform experiments in minigame of SC2 (controlling small armies) and in NeuralMMO (analogous setup).

It is worth stressing, that apart from conducting typical analysis, authors also put forward more qualitative hypotheses, that are then verified by custom experiments. I highly value this way of conducting research, as it provides way more insights into the method than just pure return plots, often encountered in similar papers.

Major comments
- Please be more explicit in the abstract, when talking about the experiments conducted. Claiming "Experimental results on StarCraft II and Neural MMO show ASN significantly improves the performance of state-of-the-art DRL approaches compared with several network architectures." could be misinterpreted as providing a solution that can play SC2 better than state-of-the-art methods, while the actual result is to be able to micromanage small fights in this environment well, and thus it would be fairer to say "on StarCraft II Minigame" or more explicitely "StarCraft II small scale fight simulation" or anything else in this spirit. These are great research domains, there is no need to overclaim the results.
- A proposed architecture assumes (to some extent) an independence of effects of action-agent pairs (action aimed at player i is conditionally independent from observation of agent j). There is still a way to represent it, but only by bypassing the proposed architecture, and expressing this behaviour directly in e^i. While this sort of decomposition makes sense for various heavily localised problems (such as micro-management of fights in games like SC2 or NeuroMMO) it looks like a potential limitation, and assumption, that is not articulated in the text. Have authors performed any analysis looking into this phenomenon?
- It is unclear, how ASN compares or relates to the architecture used in OpenAI Five network (https://d4mucfpksywv.cloudfront.net/research-covers/openai-five/network-architecture.pdf) which arguably should be used as a baseline, given that it follows a similar idea of producing per-agent embedding, that influences part of action space, that is directly related to influencing other agents (targeting opponents). Despite not being peer reviewed, it is a well known, and clearly described method, which should at the very least be acknowledged as prior work.
- Some equations are hard to parse, e.g.:
-- page 5, sum_i=1^b [(y_t^tot - Q_tot(s,a; theta))^2]  <- "i" is never used in the equation (I guess it changed meaning, and is no longer agent identity, but rather batch index?)
-- page 4, what does expectation over "t" mean? In the introduced notation, expectation is taken wrt. control policy pi_i, interpreted as a joint probability distribution over agent actions and environment state transition. Can this be unified?

Minor comments
- "When one of the arm units dies" -> "When one of the army units dies"
- details of ASN-PPO could be safely moved to the Appendix, as it is a direct substitution regular policy updates with (2), rather than a separate contribution that requires in-depth description (and given that paper is quite long, it could use a bit of message "sharpening")

I am happy to revisit (and increase) the rating assigned, once authors address the comments above, especially the relation to OpenAI Five architecture.

**Experience Assessment:**

I have published one or two papers in this area.

**Review Assessment: Checking Correctness Of Derivations And Theory:**

I assessed the sensibility of the derivations and theory.

**Review Assessment: Checking Correctness Of Experiments:**

I carefully checked the experiments.

**Review Assessment: Thoroughness In Paper Reading:**

I read the paper at least twice and used my best judgement in assessing the paper.

---

> ### Author Response · Authors · 2019-11-09
> **Answer for review #2**
>
> Thank you for your valuable and inspiring comments.
>
> [Environment] Thanks for pointing out this, we have revised the description of StarCraft II and upload our revised version.
>
> [Assume independence of effects] We do not assume the independent effects of action-agent pairs. When we calculate the Q-value or the probability of choosing an action that affects agent j, we not only consider the information about agent j, but also the environmental information and other agents’ information, which is calculated in the left part of ASN (Figure 1, orange part). Thus, our design can support cases when an action depends on multiple agent’s information. Our experimental results further confirm this as shown in Figure 6 (c). We can see the Q-value of attacking opponent 1 changes as the HPs difference of two opponents changing, and the ASN agent holds a higher Q-value of attacking opponent 1 when opponent 1's HP is lower than opponent 2 and vice versa.
>
> [Relation with openAI five] Thanks for pointing out this. We have reviewed the network structure of OpenAI Five, and the differences of OpenAI five and ASN are as follows: 1) They select action type and the target unit separately to decrease the parameter dimensions of large-scale action space, while in many cases, the selection of action type and target unit should be considered simultaneously since they are semantically intercorrelated. In contrast, our multi-action ASN determines the selection of action type and target unit simultaneously to explicitly leverage their semantic correlation. We also propose a new PS across sub-modules according to action semantics to improve training efficiency. 2) They designed network structures using the attention mechanism to determine which target unit should be selected. However, the network input arbitrarily selects the first 64 units of information from each group, which does not consider action semantics. We have added the description of OpenAI Five in the revised uploaded version as suggested.
>
> [Minor comments] Thanks for pointing out this, we have revised this and upload our revised version.

---

> > ### Comment · AnonReviewer2 · 2019-11-11
> > **Thanks for the revision and claritcation**
> >
> > I would like to thank authors for addressing the comments, and providing clarification on the independence assumption. I believe this paper is an interesting contribution, and will update my assessed score to reflect that.

---

### Official Review · AnonReviewer1 · 2019-10-23
**Official Blind Review #1**

**Rating:** 6

**Review:**

This paper focused on the multi-agent systems, and proposed a new network architecture (aka ASN). In the new architecture, the action set is  manually split into two subsets, each of which contains the actions that affects other agents or not.  Besides the net architecture, the authors also discussed several important issues including “Multi-action ASN” and  parameter-sharing. In the experiments, the authors evaluate the proposed nets in Starcraft and Neural MMO, and shows superior performance compared with other methods.

Comment:
This paper is relatively well-written and clearly introduces the basic idea of the work. The authors focuses on an important issue for MAS, i.e. how to reduce model complexity when increasing the number of agents. In their work, the authors would like to decompose the action set into two smaller sub-sets based on the so called action semantics, such that the designed nets have simpler and more straightforward (efficient) structure. It is interesting and  somehow convincing. One question on this point how to split the action set properly? My concern is from more general cases, where boundary  to classify the action set would be not so clear. In this scenario, how to obtain an proper split. And what would happen if we choose the wrong subset? At the beginning, I though the split would be learned by some sub-nets, but after reading I found it seems to manually set it. This is the motivation why I concern this problem.

Another point I concern is about the parameter-sharing (PS) issue in Page 5. I understand the authors want to impose the PS trick to reduce the training complexity for the model.  I also agree with this, but there is several points, which confuse me somewhat. (1) I’m still confused the detail of PS. Does PS means the agents share the exactly same weights, same structure, some correlation or else? (2) I know the PS trick is popularly used in multi-task learning, in which the designers have to face the question “HOW MUCH common information should the individual networks share”. It means that PS trick might results in challenging model selection problem in practice. Hence I am curious how such issue is handled in the proposed model? I know it might not be the main contribution of this paper, but I think it is important to discuss, which might be missed in the paper.

**Experience Assessment:**

I have read many papers in this area.

**Review Assessment: Checking Correctness Of Derivations And Theory:**

I did not assess the derivations or theory.

**Review Assessment: Checking Correctness Of Experiments:**

I assessed the sensibility of the experiments.

**Review Assessment: Thoroughness In Paper Reading:**

I made a quick assessment of this paper.

---

> ### Author Response · Authors · 2019-11-09
> **Answer for review #1**
>
> Thank you for your valuable and inspiring comments.
>
> [how to classify the action set] We agree with the reviewer that how to classify the action set is important. This paper can be viewed as the first step towards investigating the influence of action semantics in multiagent learning and our proposed framework of explicitly leveraging action semantics can indeed greatly improve the performance of existing approaches. Currently, the action sets are divided manually based on prior knowledge and how to automatically learn to classify the action set is worth exploring as future work.
>
> [PS] 1) Similar to previous work, we share the network weights across agents. 2) Apart from sharing the network weights across agents, we further enable PS inside each agent network. ASN divides the agent’s policy/Q network into several sub-modules according to action semantics and shares the network weights of some sub-modules. If the other agents that the agent can directly influence are homogeneous, the sub-module parameters (Figure 1, blue part) can be shared. More detailed descriptions can be found in the last paragraph of Section 3.3.

---

### Official Review · AnonReviewer3 · 2019-10-26
**Official Blind Review #3**

**Rating:** 6

**Review:**

The motivation for this paper is straightforward. The author believes that the semantic information of actions should be explicitly considered in the multi-agent reinforcement learning problem, and the information needed to make different semantic actions should be different. To this end, the author divides the action semantics into two categories, one is actions that only affect the environment and itself, and the other is actions that affects the other agents. Making a decision corresponding to the former requires relying on all local observations of the agent, but making a decision corresponding to the latter requires only local observations related to the affected agent.

To this end, the author proposes a novel network structure ASN that can achieve the above objectives. Because it is a modification of the network structure, ASN can be combined with any type of multi-agent reinforcement learning algorithm to improve its convergence speed and final performance. The author conducted a rich experiment and verified the validity and superiority of the ASN structure from various angles.

But for this paper, I have doubts in the following aspects:
1. In the experimental part, the author compares the ASN-based algorithm with QMIX and VDN in the StarCraft II environment, but the performance of the benchmark algorithm on the 8m and 2s3z maps is significantly lower than that of the original paper. Because the author did not use the RNN network when implementing these algorithms? If so, why not implement it with the RNN network? Can ASN also be combined with RNN?
2. The basic idea of ASN is similar to the attention mechanism, and the experimental part also shows that the algorithm with attention module also achieved good results. I think the author should also compare with the MAAC algorithm (Actor-Attention-Critic for Multi-Agent Reinforcement Learning, ICML2019). There are two reasons for this: First, the MAAC algorithm is a SOTA MARL method based on the attention mechanism; secondly it belongs to the actorcritic algorithm. In the comparison part of the JAL algorithm, the author only considered the value-based methods VDN, QMIX, etc., without considering the policy gradient method or the actor-critic method.
3. In the experimental part, the author also compares the proportion of ASN-based methods and benchmark algorithms that perform non-valid actions during training, such as to attack an agent outside the scope. However, in the network structure of ASN, if an agent is outside the field of view, it will not output the Q value or probability of the action corresponding to that agent. Does this mean that ASN is impossible to choose non-valid actions? So is the corresponding experiment meaningless?
4. If an action can affect multiple other agents at the same time, can the ASN network handle it?

**Experience Assessment:**

I have published one or two papers in this area.

**Review Assessment: Checking Correctness Of Derivations And Theory:**

I carefully checked the derivations and theory.

**Review Assessment: Checking Correctness Of Experiments:**

I carefully checked the experiments.

**Review Assessment: Thoroughness In Paper Reading:**

I read the paper thoroughly.

---

> ### Author Response · Authors · 2019-11-09
> **Answer for review #3**
>
> Thank you for your valuable and inspiring comments.
>
> [Performance of benchmark] We implement the benchmark algorithms with RNN (GRU layer) which can be found in our appendix. The benchmark algorithms behaving worse than the original paper is due to the fact that we remove the manual rules of forbidding agents to select illegal actions (e.g., “attack” an enemy beyond attack range or “move” beyond the border), which requires too much prior knowledge and is not practical. In contrast, our design can support automatic learning such prior knowledge and thus outperforms benchmark algorithms.
> More descriptions can also be found in the first paragraph in Section 4.1
>
> [Comparison with MAAC and other JAL policy-based, ac algorithms]: 1) their design focus is different: MAAC modifies the centralized critic network to better guide the policy network update, while ASN directly modifies an agent’s policy network according to action semantics. Thus they are not comparable directly and ASN can be incorporated into MAAC to boost its performance. We combined ASN with policy-gradient and actor-critic IL algorithms on Neural MMO to show the effectiveness of ASN, and the way to combine with JAL PG and AC methods are similar (e.g., combing ASN with MAAC).
>
> 2) Another reason we did not consider MAAC is that QMIX is the state-of-the-art algorithm [1] on StarCraft II micromanage battlefields while MAAC is not evaluated on these scenarios. We have cited and added descriptions of MAAC in the uploaded version.
>
>
> [1]. The starcraft multi-agent challenge. Samvelyan, Mikayel, et al. AAMAS, 2019.
>
> [If an agent is outside the field of view] We follow the settings of previous works that treat agents that outside of the field of view as 0-paddings input, so the output contains Q-values of actions attacking those agents. We also conduct an ablation study that replaces 0-paddings with 1-paddings and -1-paddings to show ASN can effectively extract the action semantics，and learn not to select those actions.
>
> [Handle multiple agents] We agree this is a worthwhile direction to explore as future work as mentioned in our “conclusion and future work” section. One straightforward way is to modify the input of the right part of ASN (Figure 1, blue part), that inputs the observation for a range of affected agents.

---

### Decision · Program_Chairs · 2019-12-19

**Decision:**

Accept (Poster)

**Comment:**

The authors address the challenge of sample-efficient learning in multi-agent systems. They propose a model that distinguishes actions in terms of their semantics, specifically in terms of whether they influence the acting agent and environment or whether they influence other agents. This additional structure is shown to substantially benefit learning speed when composed with a range of state of the art multi-agent RL algorithms. During the rebuttal, technical questions were well addressed and the overall quality of the paper improved. The paper provides interesting novel insights on how the proposed structure improves learning.